Iroki: automatic customization and visualization of phylogenetic trees

Moore Ryan M. 1
Harrison Amelia O. 2
McAllister Sean M. 2
Polson Shawn W. 1
Wommack K. Eric wommack@udel.edu 1
1 Center for Bioinformatics and Computational Biology, University of Delaware , Newark , DE , United States of America
2 School of Marine Science and Policy, University of Delaware , Newark , DE , United States of America
Venancio Thiago
Electronic publication date: 2020 Feb 26
Publication date: 2020
Volume: 8
Electronic Location ID: e8584
Received 2019 Oct 2; Accepted 2020 Jan 17
Copyright: ©2020 Moore et al.
Copyright year: 2020
Copyright holder: Moore et al.
License: This is an open access article distributed under the terms of the Creative Commons Attribution License, which permits unrestricted use, distribution, reproduction and adaptation in any medium and for any purpose provided that it is properly attributed. For attribution, the original author(s), title, publication source (PeerJ) and either DOI or URL of the article must be cited.
License URL: https://creativecommons.org/licenses/by/4.0/

Keywords: Bioinformatics, Environmental microbiology, Phylogeny, Sequence analysis, Metagenomics, Microbiome, Viral ecology, Microbial ecology, Data visualization, Software

Funding: USDA Agriculture and Food Research Initiative 2012-68003-30155 National Science Foundation Advances in Biological Informatics program DBI-1356374 National Science Foundation 1736030 The Established Program to Stimulate Competitive Research OIA-1736030 University of Delaware UNIDEL foundation Delaware Biotechnology Institute Delaware INBRE program National Institute of General Medical Sciences NIGMS P20 GM103446 National Institutes of Health The State of Delaware This project was supported by the Agriculture and Food Research Initiative grant no. 2012-68003-30155 from the USDA National Institute of Food and Agriculture, the National Science Foundation Advances in Biological Informatics program (award number DBI-1356374), the National Science Foundation Grant No. 1736030, the Established Program to Stimulate Competitive Research (award number OIA-1736030) from the Office of Integrated Activities, and a Doctoral Fellowship provided by University of Delaware in conjunction with the Unidel Foundation. Computational infrastructure support by the University of Delaware Center for Bioinformatics and Computational Biology Core Facility was made possible through funding from the Delaware Biotechnology Institute, and the Delaware INBRE program with a grant from the National Institute of General Medical Sciences (NIGMS P20 GM103446) from the National Institutes of Health and the State of Delaware. The funders had no role in study design, data collection and analysis, decision to publish, or preparation of the manuscript.

==============================
Phylogenetic trees are an important analytical tool for evaluating community diversity and evolutionary history. In the case of microorganisms, the decreasing cost of sequencing has enabled researchers to generate ever-larger sequence datasets, which in turn have begun to fill gaps in the evolutionary history of microbial groups. However, phylogenetic analyses of these types of datasets create complex trees that can be challenging to interpret. Scientific inferences made by visual inspection of phylogenetic trees can be simplified and enhanced by customizing various parts of the tree. Yet, manual customization is time-consuming and error prone, and programs designed to assist in batch tree customization often require programming experience or complicated file formats for annotation. Iroki, a user-friendly web interface for tree visualization, addresses these issues by providing automatic customization of large trees based on metadata contained in tab-separated text files. Iroki’s utility for exploring biological and ecological trends in sequencing data was demonstrated through a variety of microbial ecology applications in which trees with hundreds to thousands of leaf nodes were customized according to extensive collections of metadata. The Iroki web application and documentation are available at https://www.iroki.net or through the VIROME portal http://virome.dbi.udel.edu. Iroki’s source code is released under the MIT license and is available at https://github.com/mooreryan/iroki.

Introduction

Community and population ecology studies often use phylogenetic trees as a means to assess the diversity and evolutionary history of organisms. In the case of microorganisms, declining sequencing cost has enabled researchers to gather ever-larger sequence datasets from unknown microbial populations within environmental samples. While large sequence datasets have begun to fill gaps in the evolutionary history of microbial groups (Simister et al., 2012; Müller et al., 2015; Lan, Rosen & Hershberg, 2016; Larkin et al., 2016; Wu et al., 2016), they have also posed new analytical problems, as extracting meaningful trends from high dimensional datasets can be challenging. In particular, scientific inferences made by visual inspection of phylogenetic trees can be simplified and enhanced by customizing various parts of the tree.

Many solutions to this problem currently exist. Standalone tree visualization packages allowing manual or batch modification of trees are available (e.g., Archaeopteryx (Han & Zmasek, 2009), Dendroscope (Huson et al., 2007), FigTree (Rambaut, 2006), TreeGraph2 (Stöver & Müller, 2010), Treevolution (Santamaría & Therón, 2009)), but the process can be time consuming and error prone especially when dealing with trees containing many nodes. Some packages allow batch and programmatic customizations through the use of an application programming interface (API) or command line software (e.g., APE (Paradis, Claude & Strimmer, 2004), Bio::Phylo (Vos et al., 2011), Bio.Phylo (Talevich et al., 2012), ColorTree (Chen & Lercher, 2009), ETE (Huerta-Cepas, Serra & Bork, 2016), GraPhlAn (Asnicar et al., 2015), JPhyloIO (Stöver, Wiechers & Müller, 2016), phytools (Revell, 2012), treeman (Bennett, Sutton & Turvey, 2017)). While these packages are powerful, they require substantial computing expertise, which can be an impediment for some scientists. Current web based tree viewers are convenient in that they do not require the installation of additional software and provide customization and management features (e.g., Evolview (He et al., 2016), IcyTree (Vaughan, 2017), iTOL (Letunic & Bork, 2016), PhyD3 (Kreft et al., 2017), Phylemon (Sánchez et al., 2011), PhyloBot (Hanson-Smith & Johnson, 2016), Phylo.io (Robinson, Dylus & Dessimoz, 2016)), but often have complex user interfaces or complicated file formats to enable complex annotations. Iroki strikes a balance between flexibility and usability by combining visualization of trees in a clean, user-friendly web interface with powerful automatic customization based on simple, tab-separated text (mapping) files. Given its focus on automatic customization and a core set of key features, Iroki’s user interface can remain lean and easy-to-learn while still enabling complex customizations. In addition to specifying simple color gradients directly in the mapping file, Iroki also provides a dedicated module allowing the user to generate custom gradients to embed their data into color space, enhancing visualization. Iroki stays responsive even when customizing large trees, and it does not require an account or uploading potentially sensitive data to an external service.

Here, Iroki was used to customize large trees containing hundreds to thousands of leaf nodes according to extensive collections of metadata. These applications demonstrated the utility of Iroki for distilling biological and ecological insights from microbial community sequence data. The particular use cases included examinations of phage-host interactions, relative abundance of populations across sample types, and comparisons of viral community composition across environmental gradients.

Methods

Iroki is a web application for visualizing and automatically customizing taxonomic and phylogenetic trees with associated qualitative and quantitative metadata. Iroki is particularly well suited to projects in microbial ecology and those that deal with microbiome data, as these types of studies generally have rich sample-associated metadata and represent complex community structures. The Iroki web application and documentation are available at the following web address: https://www.iroki.net, or through the VIROME portal (http://virome.dbi.udel.edu) (Wommack et al., 2012). Iroki’s source code is released under the MIT license and is available on GitHub: https://github.com/mooreryan/iroki.

Implementation

Iroki is built with the Ruby on Rails web application framework. The main features of Iroki are written entirely in JavaScript allowing all data processing to be done client-side. This provides the additional benefit of eliminating the need to transfer potentially private data to an online service.

Iroki consists of two main modules: the tree viewer, which also handles customization with tab-separated text files (mapping files), and the color gradient generator, which creates mapping files to use in the tree viewer based on quantitative data (such as counts) from a tab-separated text file similar to the classic-style OTU tables exported from a JSON or hdf5 format biom file (McDonald et al., 2012).

Tree viewer

Iroki uses JavaScript and Scalable Vector Graphics (SVG, an XML-based markup language for representing vector graphics) for rendering trees. The Document Object Model (DOM) and SVG elements are manipulated with the D3.js library (Bostock, Ogievetsky & Heer, 2011). Rectangular, circular, and radial tree layouts are provided in the Iroki web application. Rectangular and circular layouts are generated using D3’s cluster layout API (d3.cluster). For radial layouts, Algorithm 1 from Bachmaier, Brandes & Schlieper (2005) was implemented in JavaScript. In addition to the SVG tree viewer, Iroki also includes an HTML5 Canvas viewer with a reduced set of features capable of displaying huge trees with millions of leaf nodes (Supplementary Materials Sec. 4).

Iroki provides the option to automatically style aspects of the tree using a tab-separated text file (mapping file). Entries in the first column of this file are matched against all leaf labels in the tree using either exact or substring matching. If a leaf name matches a row in the mapping file, the styling options specified by the remaining columns are applied to that node. Inner nodes are styled to match their descendant nodes so that if all descendant nodes moving towards the inner parts of the tree have the same style, then quick identification of clades sharing the same metadata is possible. Aspects of the tree that can be automatically styled using the mapping file include branches, leaf labels, leaf dots, bar charts, and arcs.

Inner node labels may represent support values (e.g., bootstrap results) or other comments that describe the inner nodes. If inner labels are numeric, then inner nodes can be decorated with filled and unfilled circles that allow quick identification of branches with high support. The semantics of support labels are key to proper tree representations (Czech, Huerta-Cepas & Stamatakis, 2017). As Iroki currently does not implement tree rerooting, Iroki handles these specifics implicitly rather than giving the option to map inner node labels to branches or to the nodes themselves.

While Iroki is focused mainly on automatic customization via mapping files, some interactive features are included such as node selection and the ability to modify labels after a tree has been submitted. Finally, various aspects of the tree can be adjusted directly through Iroki’s user interface.

Color gradient generator

Iroki’s color gradient generator accepts tab-separated text files (similar to the classic-style count tables exported by VIROME (Wommack et al., 2012) or QIIME 1 (Caporaso et al., 2010)) and converts the numerical data (e.g., counts/abundances) into a color gradient. Several single-, two-, and multi-color gradients are provided including cubehelix (Green, 2011) and those from ColorBrewer (Brewer, Harrower & University, 2013).

Iroki reads numerical data from tab-separated text files. Similar to the mapping file for the tree viewer, the first column should match leaf names in the tree, and the remaining columns describe whatever aspect of the data is of interest to the researcher (e.g., counts or abundance). In a dataset with M observations and N variables, the input file will then have M + 1 rows (the first row is the header) and N + 1 columns (the first column specifies observation names). From this data, Iroki can generate color gradients in a variety of ways.

Observation means

A color gradient is generated based on the mean value of each observation across all variables. In this case, an observation i would be represented as μi=∑j=1Ncij, where cij is the value of observation (row) i for variable (column) j.

Observation “evenness”

A color gradient is generated based on the “evenness” of observation i across all N variables. Then, each observation i is represented by Pielou’s evenness index (Pielou, 1966) calculated across all variables: Ei = Hi∕Hmax, where Hi is the Shannon entropy for observation i with respect to the N variables specified in the input file, and Hmax is the maximum theoretical value of Hi. In this case, Hmax occurs when observation i has equal values cij across all N variables. Thus, we calculate Pielou’s evenness index for an observation i as Ei=−∑j=1Npijlog2pijlog2N,

where N is the number of variables and pij is the proportion of observation i in variable j (i.e.,  cij∕∑j=1Ncij).

In this way, the user can map observations with high evenness (i.e., an observation with approximately the same value for each variable) to one side of the color gradient and observations with low evenness (i.e., an observation with high values in a few variables and low values in most others) to the other side of the gradient for easy identification.

Observation projection

Data reduction can be a powerful method for extracting meaningful trends in large, high-dimensional data sets. Given that microbiome or other studies in microbial ecology can have hundreds of samples and a rich set of metadata associated with those samples, data reduction often proves useful. Thus, Iroki provides a method to project the data into a single dimension and then map that projection onto a color gradient. For data reduction, Iroki conducts a principal components analysis (PCA) calculated via the singular value decomposition (SVD) using the LALOLib scientific computing library for JavaScript (Lauer, 2017). Briefly, performing singular value decomposition on the centered (and optionally scaled) matrix X, with observations as rows and variables as columns, the following decomposition is obtained: X = USVT, where the columns of US are the principal component scores, S is the diagonal matrix of singular values, and the columns of V are the principal axes. To illustrate as much variance as possible in a single dimension, the first principal coordinate is mapped onto the chosen color gradient.

Results and Discussion

Bacteriophage proteomes, taxonomy, and host phyla

Viruses are the most abundant biological entities on Earth, providing an enormous reservoir of genetic diversity, driving evolution of their hosts, influencing composition of microbial communities, and affecting global biogeochemical cycles (Suttle, 2007; Rohwer & Thurber, 2009). Due to their importance, there is a growing interest in connecting viruses with their hosts through the analysis of metagenome data. As such, researchers have used a variety of computational techniques to predict viral-host interactions including CRISPR-spacers  (Roux et al., 2016; Coutinho et al., 2017; Nishimura et al., 2017a) and tRNA matches (Bellas, Anesio & Barker, 2015; Roux et al., 2016; Coutinho et al., 2017; Nishimura et al., 2017a), sequence homology (Roux et al., 2016; Coutinho et al., 2017; Nishimura et al., 2017a), abundance correlation (Coutinho et al., 2017), and oligonucleotide profiles (Roux et al., 2015; Roux et al., 2016; Munson-McGee et al., 2018).

We used Iroki to examine phage-host interactions at the taxonomic scale by constructing a tree based on proteomic content (Rohwer & Edwards, 2002) from a subset of viral genomes from the Virus-Host DB (Mihara et al., 2016) using ViPTree (Nishimura et al., 2017b) (Fig. 1; Supplementary Materials Sec. 1). A proteomic tree clusters phage based on relationships between the collection of protein-encoding genes encoded within their genomes (Rohwer & Edwards, 2002; Nelson, 2004; Wommack et al., 2015). Specifically, ViPTree bases its clustering on normalized tBLASTx scores between genomes following the method of Mizuno et al. (2013).

Figure 1 Proteomic cladogram of viruses from Virus-Host DB.

Proteomic cladogram of viruses infecting Actinobacteria, Bacteroidetes, Cyanobacteria, Firmicutes, and Proteobacteria. Branches are colored by host phylum. Outer ring colors represent virus taxonomic family. Virus-host data is from the Virus-Host DB (Mihara et al., 2016).

Tree branches were colored by host phyla and virus family was indicated by a ring surrounding the tree using Iroki’s bar plot options (Fig. 1; Supplementary Materials Sec. 1). As shown by the branch coloring, host phyla mapped well onto the proteomic tree (i.e., large clusters of viruses that are similar in their proteomic content often infect the same host phylum). Firmicutes-infecting phage (represented by blue branches of the tree in Fig. 1) are confined almost exclusively to a large cluster in the top-left quadrant of the tree. This large cluster of mostly Firmicutes-infecting viruses can be further partitioned according to virus family, with a distinct group of myoviruses clustering separately from the other clades which include mostly siphoviruses. The Actinobacteriophage (pink) also cluster near each other with most viruses being confined to a few clusters at the bottom of the tree. The tight clustering of the Actinobacteriophage phage is likely explained by the fact that many of the viruses infect a limited number of hosts including Propionibacterium and Mycobacterium smegmatis from the SEA-PHAGES program (https://seaphages.org) (Pope et al., 2011). In contrast, the Proteobacteria-infecting viruses (green) are clustered in a few locations across the tree, with each cluster showing high levels of local proteomic similarity.

Homology and similarity-based methods have previously been shown to be effective in predicting a phage’s host (Edwards et al., 2016), perhaps because viruses that infect similar hosts are likely to have more similar genomes (Villarroel et al., 2016). Given this and the fact that the proteomic tree clusters viruses based on shared sequence content using homology and multiple sequence alignments (Rohwer & Edwards, 2002), it is unsurprising that viruses infecting hosts from the same phylum often cluster near each other on the proteomic tree. In fact, previous studies have used proteomic distance (Nishimura et al., 2017a) and other measures of genomic similarity (Villarroel et al., 2016) to transfer host annotations from viruses with known hosts to metagenome assembled viral genomes with unknown hosts. In contrast, virus taxonomy is primarily based on multiple phenotypic criteria including virion morphology, host range, and pathogenicity, rather than on genome sequence similarity (Simmonds, 2015; Simmonds et al., 2017). One study found that for prokaryotic viruses, members of the same taxonomic family (as defined by phenotypic criteria) were divergent and often not detectably homologous in genomic analysis (Aiewsakun et al., 2018). In particular, multiple viral families in the order Caudovirales were interspersed in their dendrograms. Similar results can be seen in Fig. 1, in which several Caudovirales viral families are intermixed in clusters throughout the tree.

Bacterial community diversity and prevalence of E. coli in beef cattle

Shiga toxin-producing Escherichia coli (STEC) are dangerous human pathogens that colonize the lower gastrointestinal (GI) tracts of cattle and other ruminants. STEC-contaminated beef and STEC cells shed in the feces of these animals are major sources of foodborne illness (Hancock et al., 1994; Caprioli et al., 2005). To identify possible interactions between STEC populations and the commensal cattle microbiome, a recent study examined the diversity of the bacterial community associated with beef cattle hide (Chopyk et al., 2016). Hide samples were collected over twelve weeks and SSU rRNA amplicon libraries were constructed and sequenced on the Illumina MiSeq platform (Fadrosh et al., 2014). The study found that the structure of hide bacterial communities differed between STEC-positive and STEC-negative samples.

To illustrate Iroki’s utility for exploring changes in the relative abundance of taxa in conjunction with metadata categories, a subset of cattle hide bacterial operational taxonomic units (OTUs) were selected from the aforementioned study (Supplementary Materials Sec. 2). A Mann–Whitney U test comparing OTU abundance between STEC-positive and STEC-negative samples was performed. Cluster representative sequences from any OTU with a p-value <0.2 (selected to limit the number of OTUs on the tree and to demonstrate Iroki’s features by coloring branches based on test significance) from the Mann–Whitney U test were selected and aligned against SILVA’s non-redundant, small subunit ribosomal RNA reference database (SILVA Ref NR) (Quast et al., 2012) and an approximate-maximum likelihood tree inferred using SILVA’s online Alignment, Classification and Tree (ACT) service (https://www.arb-silva.de/aligner/) (Pruesse, Glöckner & Peplies, 2012). Iroki was then used to display various aspects of the data set (Fig. 2; Supplementary Materials Sec. 2). Branches of the tree were colored based on the p-value of the Mann-Whitney U test examining change in relative abundance with STEC contamination (dark green: p ≤ 0.05, light green: 0.05 < p ≤ 0.10, and gray: p > 0.10). Additionally, bar charts representing the log of relative abundance of each OTU (inner bars) and the abundance ratio (outer bars) of OTUs in samples positive and negative for STEC are shown. The color gradient for the inner bar series was generated using Iroki’s color gradient generator. Finally, leaf labels show the order and family of the OTU and are colored by predicted OTU phylum using one of the color palettes included in Iroki.

Figure 2 Changes in OTU abundance in two sample groups.

Approximate-maximum likelihood tree of cattle hide SSU rRNA OTUs that showed differences in relative abundance between STEC-positive and STEC-negative samples. Branch and leaf dot coloring represents the p-value of a Mann–Whitney U test (dark green: p ≤ 0.05, light green: 0.05 < p ≤ 0.1, gray: p > 0.1) for changes in OTU abundance between STEC-positive samples and STEC-negative samples. Inner bar heights represent log transformed OTU abundance, and outer bars represent the abundance ratio between STEC positive and STEC negative samples (blue bars for higher abundance in STEC positive samples and brown bars for OTUs with higher abundance in STEC negative samples). Taxa labels show the predicted order and family of the OTU and are colored by the predicted phylum using the Paul Tol Muted color palette included with Iroki.

Decorating the tree in this way allows the user to explore the data and look for high-level trends. For example, Firmicutes dominates the tree (e.g., Bacillales, Lactobacillales, Clostridiales). Members of Clostridiales are at low-to-medium relative abundance compared to other OTUs on the tree. Some Clostridiales OTUs (e.g., a majority of the Ruminococcaceae) tend to be at higher abundance in STEC-positive samples, whereas other Clostridiales OTUs, namely those classified as Lachnospiraceae, tend to be at lower abundance in STEC-positive samples. Previous studies have also identified significant positive associations between STEC shedding and Clostridiales OTU abundance in general (Zhao et al., 2013) and Ruminococcus OTUs abundance more specifically (Zaheer et al., 2017). In contrast, other studies have found certain Ruminococcus OTUs associated with shedding cattle and other Ruminococcus OTUs associated with non-shedding individuals (Xu et al., 2014). Apparent contradictions may be explained by the fact that the various studies were examining the bacterial microbiome associated with different locations on the cow (e.g., GI tract, recto-anal junction, hide). In fact, significant spatial heterogeneity in community composition exists even among different sites along the gastrointestinal tract (Mao et al., 2015). Other potential explanations include methodological differences, or that variation associated with STEC presence may be better explained by using more granular groupings than taxa and OTUs (e.g., amplicon sequence variants) (Callahan, McMurdie & Holmes, 2017).

In this dataset, more of the OTUs had a higher average relative abundance (brown bars) in STEC-negative samples than in STEC-positive samples (blue bars). Similarly, in a study of the upper and lower gastrointestinal tract microbiome of cattle, a majority of differentially abundant OTUs were found to be at higher abundance in animals that were not shedding E. coli O157:H7 (Zaheer et al., 2017). In contrast, another study found that over 75% of differentially expressed OTUs were at greater abundance in STEC shedding cattle (Xu et al., 2014).

Tara Oceans viromes

The ribonucleotide reductase (RNR) gene is common within viral genomes (Dwivedi et al., 2013) and RNR polymorphism is predictive of certain biological and ecological features of viral populations (Sakowski et al., 2014; Harrison et al., 2019). As such, it can be used as a marker gene for the study of viral communities. To explore viral communities of the global ocean, we collected RNR proteins from the Tara Oceans viral metagenomes (viromes). The Tara Oceans expedition was a two-and-a-half year survey that sampled over 200 stations across the world’s oceans (Bork et al., 2015; Pesant et al., 2015). Forty-four viromes were searched for RNRs (Supplementary Materials Sec. 3). Of these, three samples contained fewer than 50 RNRs and were not used in the subsequent analysis. In total, 5,470 RNR sequences across 41 samples were aligned with MAFFT (Katoh & Standley, 2013) and post-processed manually to ensure optimal alignment quality. Then, FastTree (Price, Dehal & Arkin, 2010) was used to infer a phylogeny from the alignment. Using this tree, the unweighted UniFrac distance (Lozupone & Knight, 2005) between samples was calculated using QIIME (Caporaso et al., 2010). A tree was generated from this distance matrix in R using average-linkage hierarchical clustering. Additionally, Mantel tests identified that conductivity, oxygen, and latitude were significantly correlated (p < 0.05) with the UniFrac distance between samples (Supplementary Materials Sec. 3). Finally, Iroki was used to generate color gradients and add bar charts to visualize the data (Fig. 3). Coloring of the dendrogram with the Viridis color palette (a dark blue, teal, green, yellow sequential color scheme) was based on a 1-dimensional projection of sample conductivity, oxygen, and latitude calculated using Iroki’s color gradient generator. The color gradient generator was also used to make the color palettes used for the bar charts.

Figure 3 Tara Oceans virome similarity with associated metadata.

Average-linkage hierarchical clustering of sample UniFrac distance based on RNR sequences mined from 41 Tara Oceans viromes. Major and sub-clusters of samples (A–G) are labeled. Branch color is based on a scaled, 1-dimensional projection of sample conductivity, oxygen, and latitude onto the Viridis color gradient. Samples that are more similar to each other in branch color represent those that are more similar to each other with respect to the environmental parameters in the ordination. The first bar series (purple) represents sample conductivity (mS/cm), the second bar series (orange) represents sample dissolved oxygen levels (µmol/kg), and the third bar series (brown/green) represents sample latitude (degrees). For the first two bar series, shorter bars with lighter colors indicate lower values, while longer bars with darker colors indicate higher values. For the third series, longer, dark brown bars indicate samples with extreme negative latitudes, whereas longer, dark blue bars indicate samples with extreme positive latitudes. Samples with intermediate latitudes are represented by shorter, light colored bars. Sample labels represent the station from which the virome was acquired and are colored by sampling depth, with light blue representing surface samples and dark blue representing samples from the deep chlorophyll maximum at that station.

Coloring the dendrogram based on a projection of the environmental conditions of the samples results in samples with similar environmental metadata being similar in color. For example, the station 66 surface and deep chlorophyll maximum (DCM) samples are nearly identical to one another with respect to conductivity, oxygen, and latitude and have the same dark bluish branch color. In contrast, surface samples from stations 31 and 32 both have a lighter yellowish-green branch color. As the bar charts indicate, these two samples are very similar to one another with respect to the metadata (hence their similar coloring), but are rather different from the station 66 samples in branch color, reflecting the differences in metadata between the two groups.

The combination of dendrogram coloring and bar charts assists in finding trends in the data. Since the dendrogram is based on UniFrac distance between samples based on RNR OTUs, samples that cluster together on the tree have more similar viral communities, according to RNR gene allele content, than samples that are far from one another. In contrast, dendrogram branch coloring and the bar charts show environmental information about the samples themselves (conductivity, oxygen, and latitude). Combining these two aspects of the samples enables visualization of the relationship between the similarity of RNR-containing viral communities and the environments in which they are found.

For example, the samples in the bottom half of the tree are, in general, from northern latitudes, whereas samples towards the top tend to be from southern latitudes. In a previous study of the T4-like viral communities of Polar freshwater lakes, no significant correlation between latitude and viral community diversity was found in the Antarctic samples (Daniel et al., 2016). Though the Arctic lakes were not tested among themselves for significant associations between latitude and viral community richness (presumably due to the small latitudinal variation in Arctic sampling locations), Arctic and Antarctic lakes were tested against one another; however, no significant difference in viral diversity was seen with respect to pole of origin. The Antarctic samples from the study ranged from 67.84°S to 62.64°S, whereas the Tara Oceans viromes used to build the tree in Fig. 3 ranged from 62.18°S to 41.18°N. The increased range of samples from the Tara survey may have enabled this shift in diversity to be detected. Additionally, the previous study used g23, the gene for major capsid protein, to survey the viral community. It is possible that a functional protein like RNR is more connected with environmental conditions than a structural protein such as the T4-like major capsid protein. RNRs reduce ribonucleotides, the rate-limiting step of DNA synthesis (Kolberg et al., 2004; Ahmad et al., 2012). There are several different types of RNR, each with specific biochemical mechanisms and nutrient requirements (Nordlund & Reichard, 2006). Accordingly, the type of RNR carried by a cell or virus often reflects the environmental conditions in which DNA replication occurs (Reichard, 1993; Cotruvo & Stubbe, 2011; Sakowski et al., 2014; Srinivas et al., 2018; Harrison et al., 2019). A survey based on RNR, then, may provide more sensitivity in detecting environmental effects on viral community structure. A significant relationship between T4-like viral communities and bacterial assemblages was found however (Daniel et al., 2016), and numerous other studies have reported a significant relationship between bacterial community diversity and latitude (e.g., Ladau et al. (2013); Raes et al. (2018)). Thus, latitudinal variation in bacterial communities is likely linked to viral community variation.

Certain clusters have been marked on the tree for further analysis. Cluster A (Station 85 DCM, Station 67 surface) contains the samples with the most divergent RNR-containing viral populations (Fig. 3) according to the dendrogram. Station 85 DCM is also the sample with the lowest conductivity, highest dissolved oxygen, and most southerly latitude, suggesting that the divergent conditions of the sample with respect to the other included samples could be influencing the divergent RNR-containing viral population. Clusters B and C also offer a good point of comparison (Fig. 3). In addition to the similarity of their RNR-containing viral populations, samples in cluster B have highly similar conductivity, oxygen, and latitude (as shown by their highly similar branch color and bar charts), suggesting a close connection between sample composition and viral population. Cluster C is separate from cluster B on the dendrogram, implying their RNR-containing viral populations are less similar. The sample metadata between the two clusters is less similar as well, with Cluster B having on average a lower conductivity and higher dissolved oxygen content than samples from cluster C.

Connections between viral community composition and environment have been seen before. Salinity, which can be estimated from measurements of electrical conductivity (Pawlowicz, 2012; Pawlowicz, 2019), has been shown to affect viral-host interactions. In a viral-host system of halovirus SNJ1 with its host, Natrinema sp. J7-2, viral adsorption rates and lytic/lysogenic rates were measured at varying salt concentrations. Adsorption and lytic rate were found to increase with salt concentration, whereas the lysogenic rate decreased (Mei et al., 2015). In a system of tropical coastal lagoons, salinity was found to be one of the main factors positively affecting viral abundance (Junger et al., 2018). Viral community structure has also been associated with shifts in salinity in various environments (Bettarel et al., 2011; Emerson et al., 2013; Winter, Matthews & Suttle, 2013; Finke & Suttle, 2019). These shifts likely effect a change in the host communities, which is reflected in the shifts in viral communities.

Cluster C can be further divided into two clusters, C1 and C2. While the samples in C1 are closer to those in C2 than to those in cluster B in terms of their RNR-carrying viral populations, the samples in C1 are more similar to the samples in cluster B with respect to their metadata projection. The similar branch coloring between samples in clusters B and C1, despite their large differences in latitude, occurs because more of the variation the first principal component (the principal component on which the Viridis coloring is based) is explained by conductivity and oxygen than by latitude (Fig. 4; full ordination: Fig. S1). More striking examples can be found elsewhere in the tree. For example, station 66 surface, station 66 DCM, and station 34 surface cluster together on the dendrogram based on viral community similarity (cluster F), but the conductivity, oxygen, and latitude values for sample 34 surface are quite different from the station 66 samples. Thus, while these three metadata categories were significantly correlated with sample UniFrac distance, other factors also play a role in shaping the viral communities. Overall, using Iroki to add color and bar charts based on environmental metadata to the dendrogram based on RNR-carrying viral community structure helps visualize that high-level viral community structure can be influenced by the environmental parameters of the sample in which they originate.

Figure 4 PCA biplot of Tara Oceans virome clusters A, B, and C.

Principal components analysis biplot of Tara Oceans viromes based on sample oxygen, conductivity, and latitude. Ordination was done on all viromes, but only those from clusters A, B, and C are shown here for clarity.

Conclusions

Iroki is a web application for fast, automatic customization and visualization of large phylogenetic trees based on user specified, tab-delimited configuration files with categorical and numeric metadata. Through the use of simple configuration files, Iroki provides a convenient way to rapidly visualize and customize trees, especially in cases where the tree in question is too large to annotate manually or in studies with many trees to annotate. While Iroki includes many key features, future work is planned to increase its utility. There is no mechanism within Iroki to handle rerooting trees. As such, users must use an external program to reroot their tree before viewing it in Iroki. Customizing the tree is mainly handled by modifying the mapping file, however, Iroki could be made more interactive by allowing the user to edit certain aspects of the tree “by-hand” without having to reupload a new mapping file. Currently, Iroki allows editing leaf labels after a tree is submitted. More interactive features, such as editing label and branch styles, are planned for a future release. Finally, bringing the full feature set of Iroki’s SVG viewer to the Canvas viewer will allow users to visualize and customize huge trees quickly and easily.

Various example datasets from microbial ecology studies were analyzed to demonstrate Iroki’s utility. Iroki simplified the processes of data exploration and presentation by facilitating the mapping of various aspects of the data directly on the tree. Though these examples focused specifically on applications in microbial ecology, Iroki is applicable to any problem space with hierarchical data that can be represented in the Newick tree format.

Supplemental Information

Supplemental Information 1 Supplemental Materials and Methods

Click here for additional data file.

Figure S1 Full PCA biplot of Tara Oceans viromes

Principal components analysis biplot of 41 Tara Oceans viromes based on sample oxygen, conductivity, and latitude.

Click here for additional data file.

Figure S2 Random branch length tree

A 1,000,000 leaf tree with random branch lengths generated with rtree (using runif with default arguments for branch lengths) from the ape R package. Tree was rendered with Iroki’s Canvas tree viewer.

Click here for additional data file.

Figure S3 GreenGenes SSU rRNA tree

A collection of 331,550 full length SSU rRNA sequences from GreenGenes rendered with Iroki’s Canvas tree viewer.

Click here for additional data file.

We would like to acknowledge Barbra D. Ferrell for editing the manuscript, and the reviewers for their constructive feedback. This content is solely the responsibility of the authors and does not necessarily represent the official views of NIH.

Additional Information and Declarations

Competing Interests

Author Contributions

Data Availability

The authors declare there are no competing interests.

Ryan M. Moore conceived and designed the experiments, performed the experiments, analyzed the data, prepared figures and/or tables, authored or reviewed drafts of the paper, and approved the final draft.

Amelia O. Harrison, Sean M. McAllister, Shawn W. Polson and K. Eric Wommack conceived and designed the experiments, authored or reviewed drafts of the paper, and approved the final draft.

The following information was supplied regarding data availability:

The Iroki web app is available at https://www.iroki.net. The Iroki source code is available under the MIT license at https://github.com/mooreryan/iroki.

Data and scripts used to generate the figures are available on Zenodo: Ryan M. Moore, Amelia O. Harrison, Sean M. McAllister, Shawn W. Polson, & K. Eric Wommack. (2019). Manuscript data for “Iroki: automatic customization and visualization of phylogenetic trees” [Data set]. Zenodo. http://doi.org/10.5281/zenodo.3458510.

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
