# Peer review of "Iroki: automatic customization and visualization of phylogenetic trees"

_PeerJ, doi:10.7717/peerj.8584_

## Round 0.1 · original submission · Minor Revisions

The reviewers provided positive comments on your manuscript. Minor corrections/comments are provided below.

·

Basic reporting

Moore et al. worked in a tool for phylogenetic tree visualization and customization. The manuscript is clearly written in professional, unambiguous language. However, there is a key point that should be considered.

In the introduction section, it is unclear why the reader should use the Iroki tool instead of the usual ones. The authors say that other tools are more “complex” (lines 49-50), but this should be clearer. I suggest you to provide a table comparing the main advantages of using Iroki over others you cite in the paper (e.g. iTOL and FigTree).

Experimental design

All raw data are supplied by the authors in an organized manner to ensure reproducibility. I tested the online tool and one thing is unclear: does the software only accept rooted trees? Is it possible for the user to specify the root after submitting? I think that this point should be discussed in the paper (if applicable).

Validity of the findings

In the results section, the authors explore data on viruses and bacteria. I have some minor questions and suggestions:

1- In 2017, Czech et al published a paper about the use of support values in phylogenetic tree viewers (Mol. Biol. Evol. 34 (6): 1535–1542 - doi: 10.1093/molbev/msx055). They discuss about ambiguous semantics in tree file formats that can lead to erroneous tree views and, therefore, to incorrect interpretations of phylogenetic analyzes. I think you should do these tests to see if Iroki is free of such errors. Maybe, citing this paper would also give more strength to your work.

2 - I think Irok could be even more interactive. Can the user modify the label name in the phylogenetic tree after submitted?

3- Line 194: Why did you chose p-value < 0.2? I suggest making it clearer in the text.

Additional comments

No comment.

·

Basic reporting

I congratulate the authors for the quality of the submitted manuscript that, in my opinion, already addresses all requirements for publication. The accompanying supplementary data and online webapps and documentation links are very good, although the online documentation could contain more rendering examples.

Experimental design

The manuscript by Moore et. al presents a flexible and simple tool for visualisation of phylogenetic trees and hierarchical data and associated metadata. It very well written, presenting in a clean and straightforward manner several features of the tool. It demonstrates the tool's usefulness with three convincing examples and covers precedentes from the literature with sufficient detail.

Validity of the findings

This tool is undoubtedly useful and can only be improved as it attracts more users. I can find no errors in the example analysis presented that need fixing.

Additional comments

If the authors are considering future development of this tool, I can suggest a few features that could be added:

1) Consider letting the user name the column with leaf names to avoid forcing users to rename columns
2) If possible, more interactive tree viewers, such as interactive components from D3 or BioJS could be considered as an option for the tree canvas in use right now. This could give the web app a more dynamic look and feel
3) As mentioned above, the documentation could have more detailed rendering examples.

Keep in mind that my comments are only suggestions for future versions of the tool and should be interpreted as impediments from publication.

---

## Round 0.2 · accepted · Accept

The authors have addressed all the reviewers' criticisms. Therefore, I recommend it for publication.